# Sorafenib as an Inhibitor of RUVBL2

**DOI:** 10.3390/biom10040605

**Published:** 2020-04-14

**Authors:** Nardin Nano, Francisca Ugwu, Thiago V. Seraphim, Tangzhi Li, Gina Azer, Methvin Isaac, Michael Prakesch, Leandro R. S. Barbosa, Carlos H. I. Ramos, Alessandro Datti, Walid A. Houry

**Affiliations:** 1Department of Biochemistry, University of Toronto, Toronto, ON M5G 1M1, Canada; nardin.nano@mail.utoronto.ca (N.N.); francisca.ugwu@gmail.com (F.U.); thiago.vargasseraphim@utoronto.ca (T.V.S.); tangzhi.li@mail.utoronto.ca (T.L.); gina.abdelmalak@mail.utoronto.ca (G.A.); 2Drug Discovery Program, Ontario Institute for Cancer Research, Toronto, ON M5G 0A3, Canada; Methvin.Isaac@oicr.on.ca (M.I.); Michael.Prakesch@oicr.on.ca (M.P.); 3Institute of Physics, University of São Paulo, São Paulo SP 05508-090, Brazil; lbarbosa@if.usp.br; 4Institute of Chemistry, University of Campinas UNICAMP, Campinas SP 13083-970, Brazil; cramos@iqm.unicamp.br; 5Department of Agriculture, Food, and Environmental Sciences, University of Perugia, 06121 Perugia, Italy; alessandro.datti@unipg.it; 6Department of Chemistry, University of Toronto, Toronto, ON M5S 3H6, Canada

**Keywords:** RUVBL2, RUVBL1, Sorafenib, AAA+ proteins, Inhibition

## Abstract

RUVBL1 and RUVBL2 are highly conserved ATPases that belong to the AAA+ (ATPases Associated with various cellular Activities) superfamily and are involved in various complexes and cellular processes, several of which are closely linked to oncogenesis. The proteins were implicated in DNA damage signaling and repair, chromatin remodeling, telomerase activity, and in modulating the transcriptional activities of proto-oncogenes such as c-Myc and β-catenin. Moreover, both proteins were found to be overexpressed in several different types of cancers such as breast, lung, kidney, bladder, and leukemia. Given their various roles and strong involvement in carcinogenesis, the RUVBL proteins are considered to be novel targets for the discovery and development of therapeutic cancer drugs. Here, we describe the identification of sorafenib as a novel inhibitor of the ATPase activity of human RUVBL2. Enzyme kinetics and surface plasmon resonance experiments revealed that sorafenib is a weak, mixed non-competitive inhibitor of the protein’s ATPase activity. Size exclusion chromatography and small angle X-ray scattering data indicated that the interaction of sorafenib with RUVBL2 does not cause a significant effect on the solution conformation of the protein; however, the data suggested that the effect of sorafenib on RUVBL2 activity is mediated by the insertion domain in the protein. Sorafenib also inhibited the ATPase activity of the RUVBL1/2 complex. Hence, we propose that sorafenib could be further optimized to be a potent inhibitor of the RUVBL proteins.

## 1. Introduction

Human RUVBL1 (also known as pontin in mammalians or Rvb1 in yeast) and its paralogue human RUVBL2 (also known as reptin in mammalians or Rvb2 in yeast) share 41% sequence identity and 64% sequence similarity to each other and belong to the AAA+ (ATPases associated with diverse cellular activities) superfamily of ATPases, which is a lineage of the P-loop NTPases. This class of ATPases is present in all kingdoms of life and is divided into numerous groups, clades, and families based on structural and sequence analyses [1,2,3]. AAA+ proteins usually form oligomeric ring structures with hexameric rings being the most commonly observed and are characterized by the presence of the AAA+ module, which contains the highly conserved Walker A and Walker B motifs responsible for nucleotide-binding and hydrolysis, respectively [3,4,5]. RUVBL1 and RUVBL2 typically act as a heterohexameric complex, but also function independently [6]. Their sequences can be divided into three domains [7,8,9,10,11,12,13,14,15,16]: (i) an N-terminal αβα subdomain of the AAA+ domain (DI), (ii) an insertion domain unique to the RUVBL proteins (DII) and that is similar in structure to an oligonucleotide/oligosaccharide-binding (OB) fold, which is a ssDNA binding domain present in replication factor protein RPA, and (iii) an all α subdomain of the AAA+ domain (DIII).

RUVBL1 and RUVBL2 play diverse roles in the cell as members of several critical complexes including the chromatin remodeling complexes TIP60 and INO80 [17]. They also regulate transcriptional activities [18] and the assembly of PIKK complexes [19]. Moreover, they were found to be important in cell cycle progression through their involvement in the assembly of the telomerase complex, small nucleolar RNP complexes, RNA polymerase II complex, and mitotic spindles [20,21].

RUVBL proteins are major contributors to cellular transformation, apoptosis, cancer invasion and metastasis via their interaction with various transcription factors that are considered major carcinogenesis mediators such as c-Myc, β-catenin, and E2F1 [6]. Significantly, RUVBL2 was found to be recruited to the promoter of *KAI-1* in a complex with β-catenin resulting in the repression of KAI-1 expression, a metastasis suppressor protein, thus contributing to the enhanced invasion ability of cancer cells [22]. Increased expression of both RUVBL1 and RUVBL2 in various cancer types was reported such as hepatocellular, breast, lung, leukemia, colorectal and lymphatic carcinomas [6]. This overexpression of the RUVBLs can be used as a diagnostic tool for patients and might be predictive of how responsive patients could be to certain treatments.

Experiments showed that depleting human *RUVBL1* and *RUVBL2* with siRNA resulted in decreased cell proliferation and increased apoptosis in hepatocellular carcinoma cell lines [23]. Furthermore, depleting *RUVBL2* with siRNA in renal carcinoma cells resulted in decreased tumor cell migration and invasiveness, and increased apoptosis [24]. Therefore, the depletion of the RUVBL proteins generally results in inhibition of cell proliferation, migration and invasion making these proteins a viable target for the development of anticancer compounds.

Various studies demonstrated that the role of the RUVBL proteins in several different cancer types is dependent on their ATPase activity. For example, mutation of the Walker B (WB) motif in RUVBL1 resulted in inhibition of cellular transformation in rats [25]. Also, overexpression of RUVBL2 WB mutant repressed the function of ATF-2 [26] and inhibited cell growth in hepatocellular carcinoma cells and leukemia cells [27,28]. Silencing endogenous *RUVBL2* with siRNA resulted in increased apoptosis which was rescued with expression of *RUVBL2 WT* resistant to siRNA but was not rescued with the expression of *RUVBL2 WB* [28].

Based on the above observations, RUVBL proteins emerged as critical targets for cancer drug development for therapeutic treatments. Elkaim et al. [29] reported the first identification of inhibitors of RUVBL1 using structure-based virtual screen of 2200 molecules through molecular docking onto the ATPase binding site of the protein followed by experimental confirmation of the top 20 hits. The study resulted in the discovery of four molecules that inhibited RUVBL1 ATPase in the range of 13 to 24 µM. One of these four inhibitors was found to be a competitive inhibitor, two were found to be mixed or uncompetitive inhibitors, and one was found to be a non-competitive inhibitor [29]. In a subsequent study, the same research group synthesized four other molecules guided by their initial virtual screening results and found that only two of those molecules inhibited the ATPase activity of RUVBL1 in vitro with IC_50_ of 9 and 18 µM [30]. Only one of the two inhibitors exhibited cytotoxic effects and induced apoptosis and necrosis in cells [30]. Another group conducted an in silico screen of virtual libraries to find novel adenosine triphosphate (ATP) analogs that specifically bind to the Walker A site and that could modify RUVBL2 protein-protein interaction network [31]. Their study led to the discovery of a novel ATP mimetic called Liddean. They found that Liddean affected the oligomeric state of RUVBL2 and led to a shift of RUVBL1/2 complex localization from the cytoplasm to the nucleus in cancer cells [31].

The biotechnology company Daiichi Sankyo (Japan) filed a patent in 2015 (WIPO Patent Application WO/2015/125786) for an aminopyrazolone derivative that inhibits the ATPase activity of the RUVBL1/2 complex. They reported promising efficacy in several mouse xenograft models [32]. Also, Cleave Biosciences (CA, USA) described a compound, CB-6644, which is a derivative of the compounds described by Daiichi Sankyo that inhibited the ATPase activity of the RUVBL1/2 complex [33]. The compound showed antitumor activity when assessed in SCID-beige mice bearing human tumor xenografts derived from either Burkitt’s lymphoma (Ramos) or multiple myeloma (RPMI8226) cell lines that were among the most sensitive to CB-6644 treatment in a cell panel screen.

Both Daiichi Sankyo and Cleave Biosciences described inhibitors for the RUVBL1/2 complex only and not the individual proteins. Therefore, in this study, we concentrated on identifying inhibitors of RUVBL2, which has higher ATPase activity than RUVBL1. We performed high-throughput screening for inhibitors of the ATPase activity of human RUVBL2 that led to the discovery of sorafenib, a drug already being used in the treatment of liver and kidney cancer. Sorafenib was found to be a mixed non-competitive inhibitor of RUVBL2 with a *K_d_* value of about 22 µM. Sorafenib also inhibited RUVBL1/2 ATPase.

## 2. Materials and Methods

### 2.1. Recombinant Protein Expression and Purification

The plasmids and strains used to express and purify all the RUVBL proteins are given in Appendix A. The Profinity eXact pPAL7 expression vector is from Bio-Rad (Berkeley, CA, USA). Point mutants were generated using the QuikChange kit (Stratagen, Berkeley, CA, USA). Primers and the respective restriction cut sites are listed in Appendix A. All constructs were verified by DNA sequencing at The Centre for Applied Genomics (TCAG) facility at the Hospital for Sick Children.

To express the relevant proteins, strains were grown in Lysogeny Broth (LB) medium at 37 °C to OD_600_ = 0.6 and expression was induced with 1 mM IPTG overnight at 18 °C. Constructs with an N-terminal Profinity eXact tag were expressed in *E. coli* BL21(DE3) pRIL and purified using Profinity eXact resins according to manufacturer’s protocol. Eluted proteins were then dialyzed in buffer A (25 mM TrisHCl, pH 7.5, 50 mM NaCl, 10% glycerol, 1 mM DTT) for 4 h and then injected onto MonoQ 5/50 GL column (GE healthcare, Chicago, IL, USA) connected to either AKTA FPLC system or BioLogic DuoFlow system (Bio-Rad), and equilibrated with buffer A prior to the application of a segmented gradient from buffer A to buffer B (25 mM TrisHCl, pH 7.5, 500 mM NaCl, 10% glycerol, 1 mM DTT) over 50 mL (50 column volume). This MonoQ step was repeated once more for fractions containing the relevant protein.

N-terminal His_6_-TEV fusion constructs were expressed in *E. coli* BL21(DE3) pRIL and purified using Ni-nitrilotriacetic acid resin (NiNTA, Qiagen, Hilden, Germany) according to the manufacturer’s protocol. Eluted proteins were incubated overnight with TEV protease in a 10:1 molar ratio of protein to TEV, dialyzed in buffer C (25 mM TrisHCl, pH 7.5, 50 mM KCl, 10% glycerol, 5 mM β-mercaptoethanol) for 4 h and passed through Ni-NTA resins to separate cleaved from uncleaved proteins. Proteins were then injected onto Mono Q 10/100 GL column (GE healthcare) connected to either AKTA FPLC system (GE healthcare life sciences) or BioLogic DuoFlow system (Bio-Rad), equilibrated with buffer C with a gradient starting at buffer C to buffer D (25 mM TrisHCl, pH 7.5, 500 mM KCl, 10% glycerol, 1 mM DTT) over 120 mL (15 column volumes). The Mono Q step was repeated once more.

For co-expression of the eXact tag-RUVBL2/RUVBL1-TEV-His_6_ complex, a pCOLA-Duet1 vector encoding both eXact tag-RUVBL2 and RUVBL1-TEV-His_6_ was transformed into BL21(DE3) pRIL *E. coli*. The complex was purified first using the Profinity eXact resin and then dialyzed into buffer E (100 mM sodium phosphate, pH 7.2, 10% glycerol) for 4 h. The complex was subsequently purified using the Ni-NTA as described above, then dialyzed into buffer C and further purified using Mono Q 10/100 GL column.

All proteins were dialyzed and stored in buffer F (40 mM TrisHCl, pH 7.5, 200 mM KCl, 5 mM MgCl_2,_ 10% glycerol, 1 mM DTT). The concentrations of the purified proteins were determined by absorbance at 280 nm.

### 2.2. Analytical Size Exclusion Chromatography

200 µL of 10 µM RUVBL2 (monomeric concentration) containing 1% dimethyl sulfoxide (DMSO) and 10 µM RUVBL2 containing 1% DMSO + 30 µM sorafenib were loaded onto a Superdex 200 HR 10/30 column (GE healthcare Life Sciences) connected to an AKTA FPLC system and equilibrated with buffer G (40 mM TrisHCl, pH 7.5, 200 mM KCl, 5 mM MgCl_2_, 10% glycerol, 1 mM DTT). Elution profiles were monitored by absorbance at 280 nm at 4 °C and 1 mL fractions were collected and analyzed by SDS-PAGE. Protein molecular masses were estimated by comparison with six molecular mass standards (Sigma, St. Louis, MI, USA): thyroglobulin, 669 kDa; apoferritin, 443 kDa; β-amylase, 200 kDa; alcohol dehydrogenase, 150 kDa; bovine serum albumin, 66 kDa; and cytochrome C, 12.4 kDa.

### 2.3. ATPase Assays

The ATPase activity of the RUVBL proteins was determined using the ATP/NADH coupled ATPase assay [34]. In this assay, the regeneration of the hydrolyzed ATP is coupled to the oxidation of NADH. The ATP hydrolysis rate was determined by measuring the decrease in NADH absorbance at 340 nm in a 150 µL reaction volume. Samples were placed in 96-well flat-bottom microplates and absorbance change was monitored using SpectraMax 340PC^384^ microplate reader (Molecular Devices). Typically, the reaction consisted of 3 mM phosphoenol pyruvate, 0.2 mM NADH, 40 units/mL pyruvate kinase, 58 units/mL lactate dehydrogenase, in ATPase reaction buffer (20 mM TrisHCl, pH 7.5, 200 mM KCl, 8 mM MgCl_2_, 10% glycerol), and 5 mM ATP (or a range of ATP concentrations). The reaction components without ATP and the ATP were incubated separately at 37 °C for 10 min and then the reaction was started by adding ATP to the rest of the reaction components. The assay was performed at 37 °C and readings were taken over an hour in 20-s intervals. The rates were corrected for background signal. Rates were averaged over selected time intervals during which the absorbance decrease was linear.

For ATPase assays with chemical compounds, compounds were incubated with the protein on ice for 30 min and spun down prior to use in assay. The final compound concentration used in the assay is as specified in the figures. All molecules were dissolved in DMSO. The final amount of DMSO was 1%.

### 2.4. Analysis of the Kinetic Parameters of RUVBL ATPase

To determine the IC_50_ value for sorafenib on the ATPase activity of RUVBL2, the ATPase activity of 10 µM RUVBL2 at different sorafenib concentrations (0, 0.1, 0.2, 0.4, 0.8, 1.6, 3.2, 6.4, 12.8, 51.2, and 60 µM) was measured and the percent inhibition was obtained. The IC_50_ value was computed using the OriginPro software by fitting to the dose response function:$$y=I_{min}+\frac{I_{max}-I_{min}}{1+10^{\left(\log\left(IC_{50}\right)-\log\left[\mathrm{sorafenib}\right]\right)h}}$$
here, *y* is the measured percent inhibition, *I_min_* is the minimum percent inhibition, *I_max_* is maximum inhibition, [sorafenib] is the molar concentration of sorafenib, h is the Hill coefficient.

To obtain the kinetic parameters of RUVBL proteins, the ATPase activities of the proteins were measured at different ATP concentrations. The monomeric concentration of the protein was 10 µM, and concentration of ATP titrated was in the range of 0.1 mM to 7 mM. Each experiment was repeated in triplicate. The Michaelis-Menten *K_M_* and the maximal velocity *V_max_* were obtained by fitting the experimental initial velocity values *V*_0_ at different ATP concentrations to:$$V_{0}=\frac{V_{max}\;\;\left[S\right]}{K_{M}+[S]}$$

ATPase assay was used to measure the ATPase activity of 10 µM RUVBL2 at different sorafenib concentrations (0, 1, 3, 4, 5, and 6 µM) while titrating ATP (ranging from 0 to 6 mM). Each experiment was repeated in triplicate. *K_i_* and *K_i_′* values were calculated using the Lineweaver-Burke equation for mixed inhibition:$$\frac{1}{V_{0}}=\frac{\alpha K_{M}}{V_{max}}\;(\frac{1}{\left[S\right]}+\frac{\alpha^{\prime }}{V_{max}})$$
where *V*_0_ is the initial velocity, *V_max_* is the maximal velocity, $\alpha =1+\frac{\left[I\right]}{K_{i}}$ and $\alpha^{\prime }=1+\frac{\left[I\right]}{K_{i}^{\prime }}.$

### 2.5. ATPlite^TM^ Luminescence Assay for High-Throughput Screening

The PerkinElmer ATPlite^TM^ 1 step Luminescence ATP Detection Assay System was used to screen for inhibitors of the ATPase activity of human RUVBL2. ATPlite is an ATP monitoring system using firefly (*Photinus pyralis*) luciferase. The system is based on the detection of light produced by the reaction of ATP with luciferase and D-luciferin:$$\mathrm{ATP}\;+\;D\text{-}\mathrm{Luciferin}\;+\;O_{2}\overset{\mathrm{Luciferase}}{\underset{{\mathrm{Mg}}^{2+}}{\to }}\mathrm{Oxyluciferin}\;+\;\mathrm{AMP}\;+\;\mathrm{PPi}\;+\;CO_{2}+\;\mathrm{Light}$$

The emitted light is proportional to the ATP concentration within certain limits. The intensity of the emitted light decreases as ATP gets hydrolyzed. Typically, 10 µM RUVBL2 protomer was incubated with 100 µM of ATP at 37 °C for 3 h in the presence or absence of compounds (5 µM). RUVBL2 storage buffer F was similarly incubated with 100 µM of ATP to serve as a control. Final concentration of DMSO was 0.01%. Aliquots were taken at different time intervals from the reaction to monitor ATP hydrolysis. Reaction samples after 3 h of incubation were mixed with equal volume of ATPlite substrate and the luminescence was read after 2 min. B-scores were calculated to remove positional errors [35].

### 2.6. Surface Plasmon Resonance

SPR measurements were performed at 25 °C using a ProteOn XPR36 instrument (Bio-Rad). Samples were buffer exchanged into HEPES buffer (25 mM HEPES, pH 7.5, 200 mM KCl, 10% glycerol, 1 mM DTT) prior to the experiment. Proteins were then immobilized by amine coupling to GLH sensor chip surfaces (Bio-Rad). RUVBL2 was coupled to the chip after being diluted with acetate buffer pH 4.5 to a final concentration of 25 µg/mL. Sorafenib and its analogs, all dissolved in DMSO, were run over the chip using the interaction buffer (10 mM HEPES, pH 7.4, 150 mM NaCl, 5 mM Mg^2+^, 0.005% TWEEN 20, 3% DMSO).

For equilibrium analysis of RUVBL2-sorafenib binding, sorafenib was diluted as a series of 2-fold dilutions ranging from 150 μM to 0.29 μM in interaction buffer. Sorafenib concentrations and buffer control were injected in the analyte channels with a contact time of 120 s, dissociation time of 120 s, and a flow rate of 30 μL/min.

To perform kinetic analysis of RUVBL2-sorafenib binding, the compound was applied in a series of 2-fold dilutions ranging from 100 μM to 6.25 μM in interaction buffer. Injection of sorafenib and buffer control onto the analyte channels were performed using the following parameters: 40 s contact time, 600 s dissociation time, and 100 μL/min flow rate. Binding kinetic values for *k_on_*, *k_off_*, and *K_d_* are average values calculated by fitting each sensorgram of an SPR data set to a 1:1 Langmuir binding model using ProteOn Manager Software (Bio-Rad). Errors were derived from standard deviations of the values calculated from fitting each binding curve.

For the screen of other molecules binding to RUVBL2, sorafenib and its analogs were diluted as previously described to a final concentration of 100 μM and were injected in the analyte channels with a contact time of 120 s, dissociation time of 600 s, and a flow rate of 30 μL/min. These experiments were done in duplicates. Hit molecules that showed binding were diluted as a series of 2-fold dilutions ranging from 100 μM to 0.2 μM and injected over the analyte channels for equilibrium analysis with a contact time of 120 s, dissociation time of 180 s, and a flow rate of 30 μL/min.

All reported data were channel and double referenced, whereby ‘no ligand immobilized’ and ‘interaction buffer’ signals were both subtracted from raw data.

### 2.7. Small Angle X-ray Scattering Experiments

Small angle X-ray scattering data were collected at the Brazilian Synchrotron Light Laboratory (CNPEM-LNLS, Campinas/SP, Brazil) using a Pilatus 300 K detector (Dectris) and a monochromatic 1.488 Å wavelength X-ray beam. Sample-to-detector distance was ~1000 mm, corresponding to the q-range from 0.01 to 0.50 Å^−1^. Human RUVBL2 samples at 0.8 mg/mL in buffer G were exposed to the X-ray beam for six frames of 10 sec and one frame of 300 sec. The same was done for samples containing RUVBL2 + DMSO and RUVBL2 + 30 µM Sorafenib (in DMSO). After data inspection for X-ray damage, aggregation and interparticle interference using the ATSAS 2.7.2 package [36], averaged final curves were generated.

## 3. Results

### 3.1. Sorafenib as an Inhibitor of Human RUVBL2 ATPase Activity

As shown in Table 1, the ATPase activity (initial ATP hydrolysis rate) of human RUVBL2 is about eight-fold higher than that of human RUVBL1. If the Walker B motif in RUVBL1 or RUVBL2 is mutated by replacing the conserved aspartic acid residue (DEVH) with asparagine [RUVBL1(D302N) and RUVBL2(D299N)], then no significant ATPase activity is observed (Table 1). The RUVBL1/2 complex formed from mixing the individually purified proteins exhibited a higher ATPase activity in comparison to RUVBL1 or RUVBL2 alone. The ATPase activity obtained from mixing the individual proteins was found to be the same as that of the complex obtained from coexpressing RUVBL1 and RUVBL2 (Table 1; see Methods). To test the contributions of each of the RUVBL subunit to the ATPase activity of the complex, complexes containing one WT and one inactive protein were formed. RUVBL1/2 complex having either one of the subunits with a mutated WB motif resulted in the reduction in the ATPase activity of the complex (Table 1). Mutating the WB of RUVBL2 caused a more significant reduction in the complex ATPase activity in comparison to mutating WB of RUVBL1. RUVBL1/2 complex with WB mutations in both proteins had no significant ATPase activity (Table 1).

Based on the above results, a High-Throughput Screen (HTS) based on the ATPlite assay (see Methods) was developed to screen the DIVERSet™ collection from ChemBridge Corp. (San Diego, CA) composed of 10,000 highly diverse drug-like molecules and a small library of kinase inhibitors (200 compounds) for inhibitors of RUVBL2 ATPase activity. B-scores [35] were calculated for all the screened compounds (Figure 1A) and hits above three standard deviations were selected. 49 compounds were retested using the ATP/NADH ATPase assay (see Methods). Sorafenib and sorafenib-p-toluenesulfonate salt (Figure 1B) were identified as validated hits. As shown in Figure 1C, the ATPase activity of 10 µM of RUVBL1, RUVBL2, or RUVBL1/2 complex (protomer concentration) were measured in the presence of DMSO or 20 µM sorafenib. DMSO or sorafenib was incubated with RUVBL proteins for 30 min on ice prior to the assay. Sorafenib was able to inhibit the ATPase activity of RUVBL2 and RUVBL1/2 by about 60% and 40%, respectively. The compound had no effect on RUVBL1; however, the ATPase activity of RUVBL1 is already quite low.

We also found that sorafenib can inhibit RUVBL2 from *Saccharomyces cerevisiae* (named ScRvb2) although less efficiently (Figure 1D; 10 µM protein monomer and 30 µM sorafenib). To demonstrate that sorafenib is not a general inhibitor of AAA+ proteins, we tested the effect of the compound on the bacterial *Escherichia coli* AAA+ proteins ClpX and RavA. No significant inhibition was observed (Figure 1D; 1 µM AAA+ protein protomer and 30 µM sorafenib). These results also demonstrate that sorafenib has no effect on the ATP/NADH or ATPlite assays being used.

To identify sorafenib analogs that might be better inhibitors of RUVBL2, 17 analogs were purchased and tested (structures shown in Appendix A). None of the analogs inhibited the ATPase activity of RUVBL2 except for regorafenib (structure shown in Figure 1B). However, regorafenib was a less efficient inhibitor than sorafenib. At 15 µM regorafenib and 10 µM protein protomer concentration, regorafenib did not inhibit the ATPase activity of RUVBL1 or RUVBL1/2 and inhibited RUVBL2 by about 40% (Figure 1E).

### 3.2. Sorafenib Does Not Affect the Oligomerization of RUVBL2 nor Induces Its Aggregation

To determine whether sorafenib is inhibiting RUVBL2 ATPase activity by affecting its oligomerization, 10 µM RUVBL2 incubated with either DMSO or with 30 µM sorafenib for 30 min was analyzed by size exclusion chromatography (SEC). The SDS-PAGE gels showed that sorafenib did not have a significant effect on the elution profile of RUVBL2 (Figure 2A and Appendix A). Furthermore, since it has been reported before that sorafenib might induce the precipitation or aggregation of proteins [37], 30 µM RUVBL2 was incubated with DMSO or 87 µM sorafenib for 30 min and then spun down. No precipitation was observed. Supernatants were run on SDS-PAGE gels. The gels did not show a decrease in the levels of soluble RUVBL2 suggesting that sorafenib is not causing the precipitation of the protein (Figure 2B and Appendix A). Furthermore, the ATPase activity of RUVBL2 with increasing concentration of sorafenib was measured in the presence of Triton X-100, which should prevent or reduce potential protein aggregation. The inhibition by sorafenib still persisted under these conditions with higher sorafenib concentrations leading to lower RUVBL2 ATPase activity (Figure 2C).

Subsequently, small angle X-ray scattering (SAXS) analysis was carried out to determine if sorafenib causes gross changes in the structure or conformational equilibrium of RUVBL2. Averaged final curves of RUVBL2, RUVBL2 + DMSO and RUVBL2 + Sorafenib were overlapped and no significant differences between their scattering profiles were observed (Figure 2D, upper panel). Calculation of the radii of gyration resulted in values of 47.7 Å for apo RUVBL2, 47.9 Å for RUVBL2 + DMSO and 47.2 Å for RUVBL2 + Sorafenib. These values agree with previously observed dimensions for human RUVBL2 by X-ray crystallography (52 Å) [14]. To further investigate differences in the overall fold and flexibility of RUVBL2, dimensionless Kratky analysis was performed. Figure 2D, lower panel, shows that apo RUVBL2, RUVBL2 + DMSO and RUVBL2 + Sorafenib curves have the same Kratky profile. Taken together, these results indicate that neither DMSO nor sorafenib induce significant changes in RUVBL2 structure or oligomerization.

### 3.3. Sorafenib is a Mixed Non-Competitive Inhibitor of RUVBL2

To characterize the inhibitory effect of sorafenib on RUVBL2, the ATPase activity of the protein was measured with increasing concentration of sorafenib and the half maximal inhibitory concentration (IC_50_) of sorafenib was found to be 3.1 ± 0.8 µM (Figure 3A).

The type of inhibition caused by sorafenib on RUVBL2 ATPase activity was determined by measuring the rate of ATP hydrolysis using ATP concentrations ranging from 0.1 to 6 mM in the presence of different sorafenib concentrations (0, 1, 3, 4, 5, and 6 µM). The Lineweaver-Burk plots describing the inhibition of RUVBL2 at different inhibitor concentrations are shown in Figure 3B. Analysis of the data revealed that sorafenib acts via a mixed non-competitive inhibition mechanism with a *K_i_* of 0.9 ± 0.3 µM and *K_i_′* of 1.9 ± 0.3 µM (Figure 3B). These values are close to the IC_50_ value obtained (Figure 3A).

The binding of sorafenib to RUVBL2 was also assessed using SPR. RUVBL2 was immobilized on a sensor chip and was exposed to increasing concentrations of sorafenib (ranging from 0.29 to 150 µM in 2-fold dilutions). The SPR profile indicated an interaction between RUVBL2 and sorafenib with fast association and dissociation rates (Figure 3C). Upon equilibrium analysis using data from all different sorafenib concentrations, the dissociation constant (*K_d_*) was calculated to be 22.7 ± 3.7 µM (Figure 3C). The dissociation constant was also calculated using kinetic analysis from *k_on_* and *k_off_* values and was found to be 21.7 ± 1.0 µM, which is consistent with that calculated by equilibrium analysis (Figure 3C). Sorafenib binding to RUVBL1 was also investigated using the SPR approach; however, the calculated *K_d_* value was found to be 84.8 ± 3.7 µM by both equilibrium and kinetic analysis (data not shown). This is consistent with our finding that sorafenib does not significantly inhibit RUVBL1 (Figure 1C).

### 3.4. Effect of Sorafenib on the ATPase Activity of RUVBL Mutants

Despite the fact that RUVBL proteins are classified as AAA+ ATPases, their ATPase activity is quite low in comparison to other AAA+ proteins and even to the yeast Rvbs [38,39]. Interestingly, two aspartic acid residues present in a highly conserved motif (DLLDR; where R is the arginine finger), were found to play an important role in the ATP hydrolysis step for Rvb from the archaeon *Methanopyrus kandleri* [40]. The two aspartic acid residues and the arginine finger protrude into the ATP-binding pocket that encompasses the Walker A and Walker B motifs of the neighboring subunit. Upon mutating D to N in the DLLDR motif of archaeal Rvb, an enhancement in the ATP hydrolysis rate was observed due to a proposed decrease in the activation barrier for proton transfer from the lytic water to the closest negatively charged proton-accepting residue in WB [40].

Given that the DLLDR motif is also present in the human RUVBL2 sequence, the following single and double mutants were generated: RUVBL2 ND (D349N: DLLDR→NLLDR), RUVBL2 DN (D352N: DLLDR→DLLNR), RUVBL2 NN (D349N/D352N: DLLDR→NLLNR), and their ATPase activities were measured. RUVBL2 NN and RUVBL2 ND exhibited about 2.9- and 5.8-fold increase in the ATPase activity relative to WT, respectively, while RUVBL2 DN was almost inactive (Figure 4).

Subsequently, the effect of sorafenib on the ATPase activity of the DLLDR mutants of RUVBL2 was tested using 15 µM compound concentration and 10 µM protomer concentration. Interestingly, sorafenib inhibited the ATPase activity of all the RUVBL2 DLLDR mutants (Figure 4A,C). More interestingly, the ATPase activity of RUVBL2 mutant deleted of the insertion domain (residues E133–V238 deleted and replaced with AGA; Figure 4B), RUVBL2ΔDII, which was measured to be similar to that of RUVBL2 WT, was not inhibited by sorafenib (Figure 4). This suggests that the RUVBL2 insertion domain might influence the sorafenib binding site or that sorafenib binds directly to it, which would be consistent with the mixed noncompetitive inhibition mode of sorafenib inhibition of RUVBL2 ATPase activity (Figure 3B).

## 4. Discussion

As mentioned earlier, different groups showed that human RUVBL proteins are critical players in tumor development and metastasis. The ATPase activity of these proteins is essential for most of their roles in cancer progression. Therefore, we were interested in finding inhibitors of the RUVBL proteins and more specifically for RUVBL2 since it exhibits higher ATPase activity than RUVBL1. Our screen led us to discover sorafenib as an inhibitor of human RUVBL2.

Enzyme inhibitors can be classified as irreversible, when they bind tightly or covalently to a target protein, and as reversible, when an inhibitor can be displaced from the enzyme-inhibitor (EI) complex by, for instance, competing with the natural enzyme substrate (S) or upon dilution. Among reversible inhibitors, competitive (bind to the same site as the substrate, forming an EI complex), non-competitive (bind to a site other than the substrate binding site, forming either ESI or EI complexes) and uncompetitive inhibitors (bind to a site other than the substrate binding site, forming EI complex and blocking substrate binding) can be found [41]. Sorafenib inhibition of RUVBL2 ATPase was determined to be mixed noncompetitive (Figure 3B) suggesting that sorafenib might not directly bind to the ATPase pocket, but possibly through interaction with different motifs thus leading to conformational changes that could affect, for example, the binding of ATP to RUVBL2 or the release of ADP. Further experiments performed on RUVBL2ΔDII confirmed that the ATPase activity of RUVBL2 with truncation of its insertion domain was not inhibited by sorafenib; therefore, leading us to propose that the mechanism of action of sorafenib is mediated through its interaction with DII or that DII flexibility causes RUVBL2 to populate certain conformations to which sorafenib can bind. Recently, it was proposed that the RUVBL2 N-terminal segment may function as a lid for the nucleotide-binding site and the binding of ATP could induce the recruitment of DII by the N-terminal segment [14,16]. In this sense, binding of sorafenib to DII would influence the dynamics of DII, consequently affecting RUVBL2 nucleotide exchange rate.

Our work suggests a regulatory role of the insertion domain (DII) on the ATPase activity of human RUVBL proteins. Such a role of DII was initially highlighted in a study published in 2011, which showed that DII has an autoinhibitory function due to its flexibility since its truncation caused an enhancement of the ATPase activity of the RUVBLs [8]. However, it should be noted that in our study, we do not see such enhancement of RUVBL2ΔDII ATPase relative to that of RUVBL2 WT (Figure 4A). The difference could possibly be attributed to the methods used for protein purification, the type of the purification tag, and the presence/absence of the tag.

Sorafenib (Nexavar, BAY-43006, Bayer Pharma) is an oral drug approved by the FDA for the treatment of advanced renal cell carcinoma and hepatocellular carcinoma, and its effect on other tumor types such as breast, lung and colon was reported [42,43]. It is a multikinase inhibitor that targets the Raf serine/threonine kinase (Raf-1, WT BRAF, and oncogenic BRAF V600E) and receptor tyrosine kinases (VEGFR 1-3 and PDGFR), which explains its broad activity across different tumor types via various mechanism of actions such as antiproliferative, antiangiogenic, and proapoptotic [43]. X-ray crystal structures of sorafenib with Raf-1, WT BRAF and BRAF V600E were published and showed that sorafenib is an allosteric inhibitor that binds to the activation segment Asp-Phe-Gly (DFG) of the Raf kinase [44].

The Raf kinase exists in one of two conformations; one is the inactive state called ‘DFG Asp-out’ conformation in which the phenylalanine side chain occupies the ATP-binding pocket and aspartic acid side chain faces away from the active site [44,45]. The other one is the active conformation, which is called ‘DFG Asp-in’ conformation, in which the phenylalanine residue is rotated out of the ATP-binding pocket and the aspartic acid residue is facing into the ATP-binding pocket [44,45]. Sorafenib binds to the DFG motif and locks it in the DFG Asp-out state, thus rendering the kinase inactive [42,44]. Sorafenib has an IC_50_ value of 6 nM for Raf1 kinase and 57 nM for p38α kinase [42,46]. Our studies showed that sorafenib has an IC_50_ of 3.1 µM against RUVBL2; therefore, sorafenib is a much weaker inhibitor of RUVBL2 compared to its effect on the kinases. Hence, chemical modifications of sorafenib and further screening are needed to make sorafenib analogs that are better inhibitors of RUVBL2. Moreover, given the non-competitive mode of action, sorafenib may offer opportunities to support drug combination strategies, especially since sorafenib is an already approved FDA drug, and, thus, preclinical and clinical trials would be much easier to pursue.

## Figures and Tables

**Figure 1 biomolecules-10-00605-f001:**
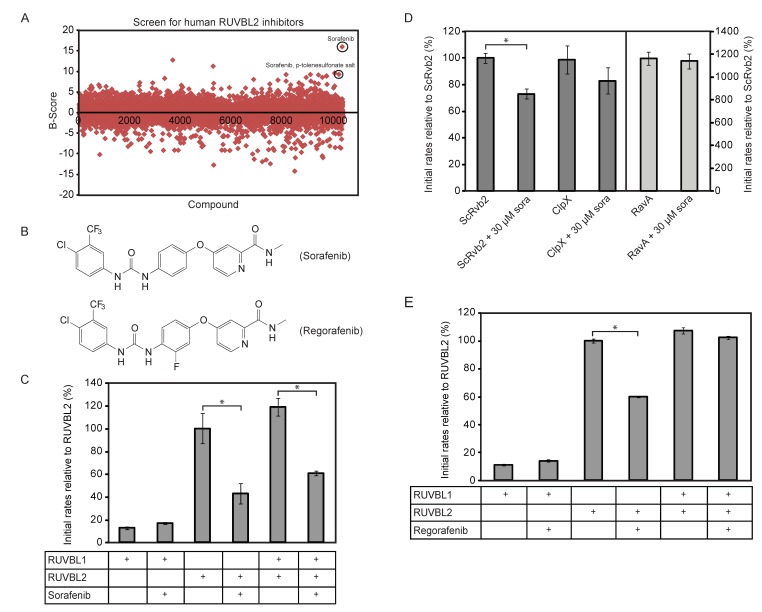
Identification of sorafenib as an inhibitor of the ATPase activity of RUVBL2. (**A**) High-Throughput Screen (HTS) based on the ATPlite assay was used to screen the DIVERSet™ collection from ChemBridge Corp. (10,000 drug-like molecules) and Kinase Inhibitor Library (200 compounds) for inhibitors of RUVBL2 ATPase activity. B-scores were calculated for all the compounds as described in the Methods section. 49 compounds were found to be above three standard deviations and were retested. Sorafenib and sorafenib-p-toluensulfonate salt were validated as hits (circled). (**B**) Shown are the structures of sorafenib (top) and its analog regorafenib (bottom). (**C**) Initial ATPase rates of RUVBL1, RUVBL2 and RUVBL1/RUVBL2 complex were measured at 10 µM protomer concentration in the presence and absence of 20 µM sorafenib. Initial rates of proteins are reported relative to that of RUVBL2 WT as percentage. (**D**) Initial ATP hydrolysis rates of yeast Rvb2 (ScRvb2; 10 µM protomer concentration), *E. coli* ClpX (1 µM protomer concentration) and *E. coli* RavA (1 µM protomer concentration) were measured in the presence and absence of sorafenib at 30 µM concentration. Initial rates are reported relative to that of ScRvb2 as percentage. (**E**) Initial ATP hydrolysis rates of 10 µM protomer concentration of RUVBL1, RUVBL2, and RUVBL1/RUVBL2 complex were measured in the presence and absence of 15 µM regorafenib. A (*) in panels (**C**–**E**) indicates that the difference between the two compared ATPase activities is significant with *p*-value < 0.01.

**Figure 2 biomolecules-10-00605-f002:**
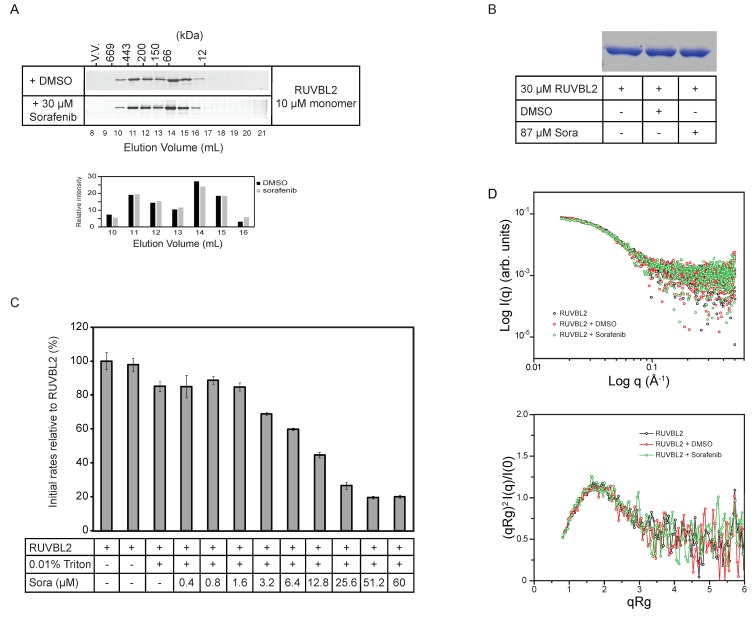
Effect of sorafenib on RUVBL2 oligomerization and aggregation. (**A**) 10 µM RUVBL2 was incubated with DMSO or 30 µM sorafenib and then loaded onto gel filtration column (Superdex 200) at 4 °C. 1 mL fractions were collected, and proteins were visualized on SDS-PAGE gel by silver staining. Proteins were incubated with DMSO or sorafenib for 30 min on ice prior to loading on the column. The positions of the molecular mass standards in kilodaltons are given along the upper x-axis. The bar graph below the gels shows the quantifications of the bands in elution volumes 10 to 16 mL. (**B**) SDS-PAGE gel showing RUVBL2 alone, with DMSO or with sorafenib. 30 µM RUVBL2 was incubated with DMSO or with 87 µM sorafenib on ice for 30 min. Samples were spun down and then supernatants were loaded on the gel. Gel image shows that sorafenib does not cause aggregation of RUVBL2 at the tested concentrations. (**C**) ATPase activity of RUVBL2 (10 µM protomer concentration) with increasing concentrations of sorafenib in the presence of 0.01% Triton. Bar graph shows decreasing activity of RUVBL2 with increasing concentration of sorafenib even in presence of Triton which should prevent protein aggregation. (**D**) Small angle X-ray scattering data of human RUVBL2. Scattering profiles (upper panel) and dimensionless Kratky plot (lower panel) of apo RUVBL2, RUVBL2 + DMSO and RUVBL2 + sorafenib. The overlap of the curves suggests that, overall, sorafenib does not induce significant structural changes in RUVBL2 nor affects the protein’s oligomeric state.

**Figure 3 biomolecules-10-00605-f003:**
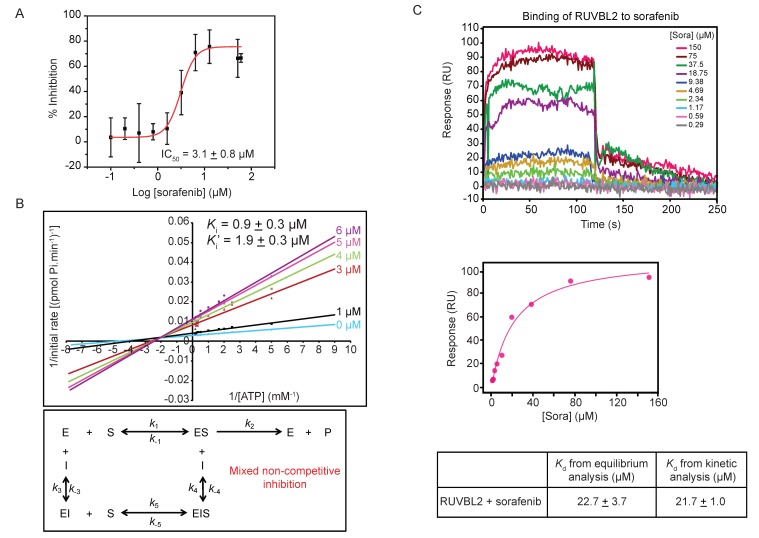
Mechanism of sorafenib inhibition of RUVBL2 ATPase activity. (**A**) Determination of IC_50_ of sorafenib inhibitor using ATP/NADH coupled ATPase assay with 10 µM RUVBL2, 5 mM ATP, and a range of sorafenib concentrations (0, 0.1, 0.2, 0.4, 0.8, 1.6, 3.2, 6.4, 12.8, 51.2, and 60 µM). The IC_50_ was computed using the OriginPro software. The graph shows the mean ± SD of one experiment of triplicates. (**B**) Top panel shows double reciprocal plot (Lineweaver-Burk plot) of the reaction rate with 10 µM RUVBL2, range of ATP concentrations (0.1 to 6 mM), and increasing concentrations of sorafenib (0, 1, 3, 4, 5, and 6 µM). Each line in the graph is the mean of one experiment done in triplicates. *K_i_* and *K_i_′* values are reported at the top. Bottom panel shows the inhibition mechanism deduced from the pattern of Lineweaver-Burk plot. (**C**) Determination of binding constant of sorafenib to RUVBL2 using SPR. Top panel shows SPR sensorgrams of immobilized RUVBL2 WT on GLH sensor chip exposed to ten 2-fold serial dilutions of sorafenib in triplicate. SPR sensorgrams for injected sorafenib are shown as colored lines and are channel and double referenced (subtracting both interaction buffer signal and no ligand immobilized signal). Middle panel shows equilibrium analysis of RUVBL2-sorafenib binding. The bottom panel shows *K_d_* values calculated from equilibrium and kinetic analysis of the SPR data.

**Figure 4 biomolecules-10-00605-f004:**
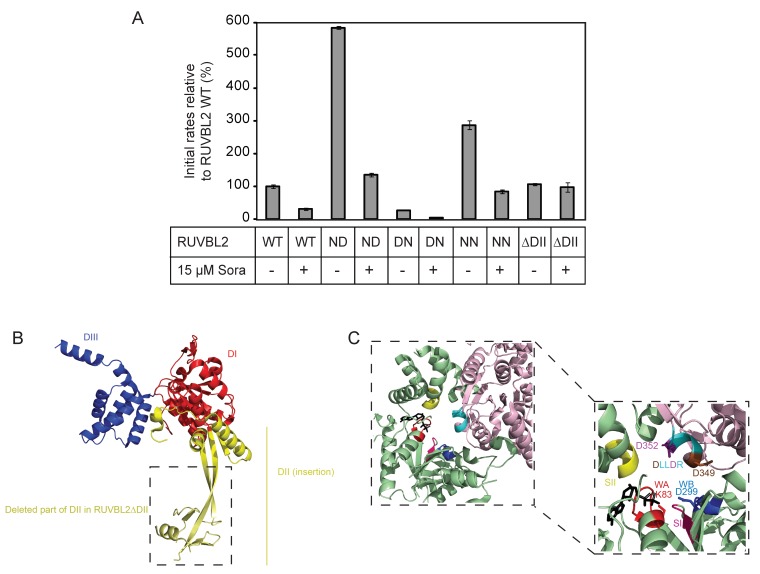
Effect of sorafenib on the ATPase activity of RUVBL2 mutants. (**A**) ATPase activities of RUVBL2 WT, with mutations in the DLLDR motif, and with truncation of the DII domain were measured at 10 µM protomer concentration in the absence and presence of 15 µM sorafenib. Results shown are the mean of triplicates ± SD. The following are the mutants used, ND: NLLDR; DN: DLLNR; NN: NLLNR. (**B**) Crystal structure of RUVBL2 monomer (PDB: 6H7X). In red is DI, the N-terminal αβα subdomain of the AAA+ domain, in blue is DIII, the C-terminal all α subdomain of the AAA+ domain, and in yellow is DII, the insertion domain. Boxed is the part of DII that was deleted in RUVBL2ΔDII. (**C**) Shown is the ATP-binding pocket at the interface between two RUVBL2 subunits. One subunit is colored in light pink and the adjacent monomer is colored in green (PDB: 3UK6); the ADP molecule is colored in black. Conserved motifs and specific amino acids with significant role in ATP-binding and hydrolysis are colored as follows: Walker A (WA) in red with K83 in stick representation, Walker B (WB) in dark blue with D299 in stick representation, Sensor I in hot pink, Sensor II in yellow, and the DLLDR motif in cyan. The first aspartic acid (D349) in DLLDR is colored in brown and the second aspartic acid (D352) is colored in purple and both are shown in stick representation.

**Table 1 biomolecules-10-00605-t001:** Initial rates for the ATPase activity of different RUVBL proteins.

	Initial Rate of ATPase Activity [(pmol Pi) min^−1^ (pmol RUVBL Protomer)^−1^]
RUVBL1	0.015 ± 0.001
RUVBL1 + sorafenib	0.020 ± 0.001
RUVBL1 WB	0.009 + 0.004
RUVBL2	0.119 ± 0.016
RUVBL2 + sorafenib	0.051 ± 0.011
RUVBL2 WB	0.015 + 0.008
RUVBL1/2 mixed	0.142 ± 0.009
RUVBL1/2 mixed + sorafenib	0.072 ± 0.002
RUVBL1/2 coexpressed	0.132 ± 0.008
RUVBL1 WT + RUVBL2 WB	0.009 + 0.002
RUVBL1 WB + RUVBL2 WT	0.081 + 0.006
RUVBL1 WB + RUVBL2 WB	0.012 + 0.004

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
