# Peer review of "Sorafenib as an Inhibitor of RUVBL2"

_biomolecules, 2020, doi:10.3390/biom10040605_

Round 1

Reviewer 1 Report

The paper by Walid Houry and colleagues show sorafenib could inhibit RUVBL2 ATPase activity with micromolar concentration. They show data to suggest that it does not affect oligomerisation and gross conformation. Data suggest it is a mixed non-competitive inhibitor that might interact RUVBL2 DII domain. Most of the studies in this report were performed using RUVBL2 protein which can form hexamers on its own although RUVBL1-RUVBL2 heterohexamer are found in functional complexes. One of the rational is that the ATPase activities of RUVBL2 and RUVBL1-RUVBL2 are similar and so do the inhibitory effects of sorafenib. However, in RUVBL1-RUVBL2 hexamer, there are only 3 RUVBL2 protomers while there are 6 in RUVBL2 hexamer, suggesting RUVBL1 plays a role in regulating RUVBL2 ATPase in the mixed hexamer. Therefore subsequent studies ideally, although not strictly require, should be performed on the mixed hexamer as they are more physiologically relevant, especially if the inhibitor is acting non-competitively.

Some minor points:

  • When introducing RUVBL D II, the authors should point out that it is a OB fold, normally found in ssDNA binding.
  • SEC profiles cannot assess if the oligomerisation state is more or less stable.
  • SAXS does not have insufficient resolution to distinguish conformational changes which were typically observed for AAA+ ATPases upon nucleotide binding.
  • Please explain the type of inhibitors and discuss the pros and cons.
  • The effects on DII deletion needs to be toned down as contrary to other reported data, deletion of DII has no effects on the ATPase activity in the authors assays. Thus it casts some uncertainty on the data.
  • Further one can’t rule out that sorafenib affects movement of DII indirectly.

Author Response

  1. Therefore subsequent studies ideally, although not strictly require, should be performed on the mixed hexamer as they are more physiologically relevant, especially if the inhibitor is acting non-competitively.

We do indeed show the effect of sorafenib on the RUVBL1/2 complex. This is shown in Figure 1C and Table 1. We also mention this in the abstract.

  1. When introducing RUVBL D II, the authors should point out that it is a OB fold, normally found in ssDNA binding.

We explain this now on page 3.

“…(ii) an insertion domain unique to the RUVBL proteins (DII) and that is similar in structure to an oligonucleotide/oligosaccharide-binding (OB) fold, which is a ssDNA binding domain of the replication factor RPA,..”

  1. SEC profiles cannot assess if the oligomerisation state is more or less stable. SAXS does not have insufficient resolution to distinguish conformational changes which were typically observed for AAA+ ATPases upon nucleotide binding.

We agree with the reviewer, and we have modified the text accordingly. SEC and SAXS are only used to see of there are any significant changes to the oligomerization or conformation of the protein.

  1. The effects on DII deletion needs to be toned down as contrary to other reported data, deletion of DII has no effects on the ATPase activity in the authors assays. Thus it casts some uncertainty on the data. Further one can’t rule out that sorafenib affects movement of DII indirectly.

We raise this issue in the discussion. We think that the previous study did not correctly measure their ATPase activity. On page 12, we state

“Our work suggests a regulatory role of the insertion domain (DII) on the ATPase activity of human RUVBL proteins. Such a role of DII was initially highlighted in a study published in 2011, which showed that DII has an autoinhibitory function due to its flexibility since its truncation caused an enhancement of the ATPase activity of the RUVBLs [8]. However, it should be noted that, in our study, we do not see such enhancement of RUVBL2ΔDII ATPase relative to that of RUVBL2 WT (Figure 4). The difference could possibly be attributed to the methods used for protein purification, the type of the purification tag, and the presence/absence of the tag.”

Reviewer 2 Report

The manuscript entitled “ Sorafenib as an inhibitor of RUVBL2” is very well designed and written. It is mainly focused on sorafenib activity on RUVBL1 and RUVBL2 ATPase. I am impressed by the outcomes of studies, as well as by the methods used.

I have only minor concerns:

  • The abstract could be improved to correspond better with the manuscript. My advice is to shorten the enzymes characterisation and to extent the description of methods and results. I have the impression that the statement “herein we describe a high-throughput screen for inhibitors of the ATPase activity of human RUVBL2” is not the main purpose of this study.
  • Please complete key words.
  • pp 12 lines 406-408: The equation describing the reaction of ATP with luciferase should be corrected.

I recommend manuscript for publication in Biomolecules after addressing the above mentioned concerns.

Author Response

  1. The abstract could be improved to correspond better with the manuscript. My advice is to shorten the enzymes characterisation and to extent the description of methods and results. I have the impression that the statement “herein we describe a high-throughput screen for inhibitors of the ATPase activity of human RUVBL2” is not the main purpose of this study.

We have modified the abstract accordingly.

  1. Please complete key words.

Key words have been added

  1. pp 12 lines 406-408: The equation describing the reaction of ATP with luciferase should be corrected.

We are not sure what the reviewer is referring to. In any case, we have modified the equation to be the same as that given by PerkinElmer.